# *Salmonella* Control in Fattening Pigs through the Use of Esterified Formic Acid in Drinking Water Shortly before Slaughter

**DOI:** 10.3390/ani13182814

**Published:** 2023-09-05

**Authors:** María Bernad-Roche, Clara María Marín-Alcalá, Juan Pablo Vico, Raúl Carlos Mainar-Jaime

**Affiliations:** 1Departamento de Patología Animal, Facultad de Veterinaria, Instituto Agroalimentario de Aragón-IA2, Universidad de Zaragoza-CITA, 50013 Zaragoza, Spain; mbernadroche@gmail.com; 2Departamento de Ciencia Animal, Centro de Investigación y Tecnología Agroalimentaria de Aragón, Instituto Agroalimentario de Aragón-IA2, Universidad de Zaragoza-CITA, 50059 Zaragoza, Spain; cmarin@unizar.es; 3IRNASUS-CONICET-Universidad Católica de Córdoba, Facultad de Ciencias Agropecuarias, Universidad Católica de Córdoba, Córdoba 5000, Argentina; juanpablo.vico@ucc.edu.ar

**Keywords:** *Salmonella*, pigs, formic acid, water, control, microbial load, slaughter

## Abstract

**Simple Summary:**

Salmonellosis is a public health concern, and *Salmonella*-shedding pigs at the abattoir are one of the main sources of human infection. In this study, an esterified formic acid was applied as an on-farm treatment at a dose of 10 kg of product per 1000 L of water five days before slaughter. It was found that it could significantly reduce the proportion of *Salmonella*-shedding pigs arriving at the slaughterhouse and the *Salmonella* loads in the guts of the shedder pigs. These promising results suggest that this esterified formic acid could be used in farm drinking water a few days before slaughter as a complementary mitigation strategy for *Salmonella* control.

**Abstract:**

The presence of *Salmonella* in pig feces is a major source of abattoir and carcass contamination, and one of the main sources of human salmonellosis. This study assessed whether using a form of esterified formic acid (30% formic acid) in drinking water (10 kg/1000 L) 5 days before slaughter could be a helpful strategy to mitigate this public health issue. Thus, 240 pigs from three *Salmonella*-positive commercial fattening farms were selected. From each farm, 40 pigs were allocated to a control group (CG) and 40 to a treatment group (TG). At the abattoir, fecal samples from both groups were collected for *Salmonella* detection (ISO 6579-1:2017) and quantification (ISO/TS 6579-2:2012). *Salmonella* was present in 35% (95% IC = 29.24–41.23) of the samples collected. The prevalence was significantly higher in the CG than in the TG (50% vs. 20%; *p* < 0.001). In all farms, the TG showed a lower percentage of shedders than the CG. A random-effects logistic model showed that the odds of shedding *Salmonella* were 5.63 times higher (95% CI = 2.92–10.8) for the CG than for the TG. Thus, the proportion of pigs shedding *Salmonella* that was prevented in the TG due to the use of this form of organic acid was 82.2%. In addition, a *Chi*-squared analysis for trends showed that the higher the *Salmonella* count, the higher the odds of the sample belonging to the CG. These results suggest that adding this type of acid to drinking water 5 days before slaughter could reduce the proportion of *Salmonella*-shedding pigs and the *Salmonella* loads in the guts of shedder pigs.

## 1. Introduction

*Salmonella* infection is a major cause of foodborne outbreaks in the European Union and the second most frequently reported zoonosis in humans [1]. Although most *Salmonella*-infected pigs may not display any symptoms, they act as carriers of the infection, and contaminated pork and product thereof are among the most common sources of human infection [1].

Pork usually becomes contaminated at the abattoir, and the primary source of the contamination of abattoirs is the presence of *Salmonella* in the feces of slaughter pigs [2,3]. The proportion of slaughter pigs showing high loads of *Salmonella* in the cecal content appears to be directly related to the number of contaminated carcasses in the abattoir [4]. Therefore, reducing the number of pigs shedding *Salmonella*, and/or their intestinal loads of these bacteria, at the time they arrive to the abattoir can be foreseen as an important step to reduce overall abattoir environmental contamination and the proportion of contaminated carcasses.

Most shedding pigs become infected when still on the farm; thus, different approaches have been suggested to reduce the transmission of *Salmonella* infection within farms. For instance, enhancing farm biosecurity and improving cleaning and disinfection are considered some of the most important measures to control *Salmonella* in indoor settings [5,6]. However, proper levels of biosecurity will not necessarily reduce farm *Salmonella* prevalence [7,8,9], likely because its routine implementation and maintenance depend upon the farmers’ perception of the problem [10,11], and pig salmonellosis is considered of low concern for pig producers.

Vaccination has also been proposed for on-farm *Salmonella* control, and there are some commercially available vaccines against *Salmonella* infection in pigs [12]. Results from vaccine studies suggest that they may help to somehow reduce *Salmonella* shedding from specific serotypes [13], but cross-protection against different serotypes is unlikely [12]. Since pigs may be infected by many of them [14], vaccination alone may not be enough. 

Another general on-farm strategy against *Salmonella* is the administration of organic acids (OAs) through feed or water. However, their efficacy against *Salmonella* has been found to be very variable, with a number of studies showing some positive results [15,16,17,18,19,20,21,22,23,24] but others not observing any effect [25,26,27]. This variability may be attributed to several factors, such as the type, form, dose, and duration of the treatment applied; the pig’s age [28]; or even infections occurring during transport or lairage in the slaughterhouse [29]. 

The efficacy of OAs is based on their presence, in appropriate amounts, along the gastrointestinal tract, particularly in its distal parts (the cecum and large intestine), where *Salmonella* is more commonly found [30,31,32]. Lowering the pH in both the extra- and intra-cellular environment is the main mechanism by which OAs prevent *Salmonella* viability [33,34]. However, as mentioned before, most fattening pigs usually become infected with *Salmonella* while on the farm and, to a lesser extent, during transport or lairage [35,36]. After infection, pigs may shed large amounts of bacteria for a few days, after which *Salmonella* may remain silent within tonsils and lymph nodes, with no shedding until after a stressful situation occurs, such as the transport to the abattoir or while pigs are held in lairage [37,38,39]. Since a 12–18 h fasting is usually required before slaughtering, any in-feed treatment may not be effective to control *Salmonella* shedding when pigs arrive at the abattoir, as not enough OA is left within the intestine. In-feed treatments may then help to reduce the level of shedding while pigs are on the farm, but they may not prevent shedding during transport or at the slaughter of pigs already infected on the farm [21]. 

On the contrary, administering OA through farm water may have the advantage that pigs would have access to OA even during fasting, thus receiving the treatment until the moment that they are loaded onto the truck. Thus, depending upon the duration of the transport, treatment through drinking water could prolong the presence of OA in the gastrointestinal tract almost until the time of slaughter. However, as noted earlier, the effectiveness of OA treatment in this way may also vary depending on various factors. To optimize the effectiveness of OA treatment through water, researchers have investigated different types, forms, and doses of OA. Short- and medium-chain fatty acid glycerides combined with glycerol present an amphipathic structure that makes them soluble in water while maintaining their in vitro and in vivo efficacy against Gram-negative bacteria [40]. In addition, due to their glyceride form, these OAs are odorless and non-corrosive. 

A previous study suggested that the esterified form of formic acid could be effective in decreasing the prevalence of pigs shedding *Salmonella* at slaughter when administered in drinking water during lairage at the abattoir. Although the treatment was applied overnight and for only a few hours before slaughter, a significant reduction in the prevalence of shedders in the TG was observed (16.4%; [41]). This finding suggests that on-farm water treatment for a longer time with this esterified OA could be a useful strategy to decrease the prevalence of *Salmonella* shedding at slaughter. Thus, this research aimed to investigate the efficacy of this OA when administered in drinking water a few days before slaughter as a potential strategy for reducing the number of pigs shedding *Salmonella* at slaughter and the *Salmonella* loads in the gastrointestinal tract of fattening pigs. 

## 2. Materials and Methods

### 2.1. Experimental Design and Farm Sampling

Three replicates of a field trial were carried out between May and November 2022 on three different commercial pig farms located in the NE of Spain. Each farm was composed of different fattening units (average size of 1000 animals/fattening unit), two of which were selected for this study. One was assigned as the treatment group (TG) and the other as the control group (CG). Within each farm, the selection of the two fattening units to be included was based on the following criteria: (i) animals came from the same sow herd and had a similar age (less than a one-week difference) and (ii) both fattening units were *Salmonella*-positive. Within a farm, the TG was always assigned to the fattening unit with the highest seroprevalence.

To confirm the presence of *Salmonella* in the fattening units and to assess their *Salmonella* status before starting the trial, pooled floor fecal samples (PFFs) from 10 representative pens and 30 individual blood samples from pigs distributed along the fattening unit (1 pig per pen) were collected in each unit approximately one month before slaughter. 

### 2.2. Product Used

A formic acid (30% formic acid) esterified in the form of mono-, di-, and tri-glyceride with glycerol (MOLI-M C1, Molimen SL, Barcelona, Spain) was selected for the treatment group. It was applied at a dose of 10 kg of product per 1000 L of farm drinking water (EMEC AMS PLUS dosing pump, EMEC, Fitchburg, MA, US). This dose of product (10 g/kg) was below the maximum threshold value of 12 g/kg formic acid approved for the use in pigs in the European Union [42]. The treatment began 5 days before slaughter. Pigs fasted for an average of 18 h before being transported to the abattoir, but the treatment in the drinking water supply was continued until the moment that they were loaded onto the truck.

### 2.3. Slaughter Sampling

Animals from each group (CG and TG) were slaughtered separately, always beginning with the CG on one day and the TG one or two days later. The farms were less than 2 h from the abattoir, and all animals were transported on the day of slaughter. At the slaughter line, individual gastrointestinal tracts were collected from 40 pigs from each group. They were selected among all the pigs sent for slaughter in that batch (around 200 pigs), following a systematic sampling (one every three–four pigs) and avoiding any pig showing intestinal rupture during evisceration. Intestinal (colon) content (IC) samples were further collected and submitted to the laboratory for immediate processing. 

### 2.4. Isolation and Quantification of Salmonella

*Salmonella* isolation from the PFF and IC samples was carried out according to the standard ISO 6579-1:2017 method [43]. The quantification of *Salmonella* was performed on 10 IC samples selected randomly per group and per trial (except for farm 1, where only 9 were analyzed due to 1 being inadvertently forgotten) following the miniaturized Most Probable Number (MPN) enumeration method (ISO/TS 6579-2:2012 [44]). Briefly, in a 4 × 3-well microtiter plate, 2.5 mL of the initial 1:10 buffered peptone water (BPW) suspension used for isolation was transferred to each of the three wells of the first column. Then, 500 µL from each well was transferred and mixed with 2 mL of BPW in the wells from the second column. Dilutions were performed in the same way in the third and fourth columns making a total of 4 dilutions (1:10, 1:50, 1:250, 1:1250) for each sample to be analyzed. Each sample was analyzed in triplicate. Dilutions were incubated at 37 °C ± 1 °C for 18 h ± 2 h. Then, 20 µL of each BPW dilution was transferred onto MRSV plates and incubated at 41.5 °C ± 1 °C for 24 h ± 3 h for selective enrichment. Negative wells after 24 h were further incubated for up to 48 h. Suspect wells of (at least) the highest dilution(s) were plated out on selective isolation medium XLD at 37 °C ± 1 °C for 24 h ± 3 h. From each plate, a characteristic colony was biochemically tested (indole reaction, urea and triple sugar iron agar tests, and L-lysine decarboxylation) (Panreac Química SAU, Castellar del Vallés, Spain) for definitive confirmation. The MPN characteristic number was obtained by counting the number of positive wells in the 4 dilution × 3 repetition system used. Then, MPN was generated using the MPN calculation program version 6 [45]. The results are expressed as MPN per mL or g. The detection limit of the mini-MSRV method is approximately 1 CFU/g.

### 2.5. Serological Analysis

The Herdcheck Swine *Salmonella* ELISA test (IDEXX Laboratories, Westbrook, ME, USA) was employed according to the manufacturer’s guidelines to detect specific antibodies (IgG) directed against *Salmonella.* For seroprevalence estimates, and due to the test’s limited sensitivity and specificity on field samples (73% and 95%, respectively; [46]), a cut-off value of OD% ≥ 40 was used to deem a pig seropositive.

### 2.6. Statistical Analyses

To assess the efficacy of the treatment in each trial, Fischer’s exact test was used to compare the prevalence of *Salmonella* shedding at slaughter between the CG and the TG. Further, a random-effects logistic regression analysis, considering the results from all the trials together, was carried out to obtain an overall estimate of the effect of the treatment on the prevalence of *Salmonella* shedding at slaughter. For this purpose, the presence of *Salmonella* in the IC samples was considered as the dependent variable and the treatment as the independent one. The farm was considered as the random (grouping) variable to account for the likely correlation among individuals within the same farm. The Odds Ratio (OR) between the CG and TG and its corresponding 95% confidence interval (95% CI) were estimated, and from these, an estimate of the proportion of pigs that may have been prevented from shedding because of the use of the OA, that is, the attributable fraction (AF), was calculated as OR-1/OR [47].

The Most Probable Number results were categorized into four groups: (1) ND, not detected (below the limit of detection); (2) between 0.02 and 20.00 CFU/g; (3) between 21 and 200; and (4) exceeding the limits for counting (∞). A *Chi-square* analysis for linear trends was applied to compare the MPN categories between the CG and TG for all the trials together. These statistical analyses were performed using STATA software (STATA/IC 12.1. Stata-Corp. LP, College Station, TX, USA). 

## 3. Results

### 3.1. Farm Salmonella Status Prior to the Trial

One month before starting the trials, the *Salmonella* status of all the farms was assessed, and the fattening units were selected. In all of the units included in this study, *Salmonella* was detected, with all presenting 10% of pens positive for *Salmonella* (Table 1). With regard to the serological results, the farms’ overall seroprevalences (considering the two selected fattening units) were 21.6% in farm 1, 38.3% in farm 2, and 48.3% in farm 3. In two farms (1 and 2), no significant differences in seroprevalence were found between the two selected fattening units. In farm 3, however, in one of the units, the seroprevalence was significantly higher than that of the other one (63.3% vs. 33.3%) (Table 1). 

### 3.2. Salmonella Shedding at Slaughter

In farm 1, a high proportion of *Salmonella* shedders was observed in both groups, but this proportion was significantly higher in the CG than in the TG (65% vs. 27.5%, respectively; *p* = 0.0015) (Table 2). The odds of *Salmonella* shedding in the CG were almost 5 times higher than in the TG (OR = 4.9; 95% CI = 1.9–12.7). The MPN results showed higher counts of *Salmonella* in the CG than in the TG. *Salmonella* was not detected in 88.9% (eight out of nine) of the samples from the TG or in 55.6% (five out of nine) of the samples from the CG, while there was one sample exceeding the limits for counting in both groups (Table 3). 

In farm 2, an overall low *Salmonella*-shedding prevalence was observed. *Salmonella* was not detected in the TG, while the CG showed few pigs shedding *Salmonella* (0% vs. 12.5%; *p* = 0.055) (Table 2). Regarding the MPN results, in the CG, only one positive *Salmonella* sample was detected, with low counts (3.8 MPN/g). No *Salmonella*-positive samples were detected in the TG (Table 3).

In farm 3, a high *Salmonella*-shedding prevalence was observed in both groups. More shedder pigs were found in the CG than in the TG (72.5% vs. 32.5%, respectively; *p* = 0.0007) (Table 2). The odds of *Salmonella* shedding in the control group were more than 5 times higher than in the TG (OR = 5.5; 95% CI = 2.1–14.3). The MPN results showed higher counts of *Salmonella* in the CG than in the TG. *Salmonella* was not detected in 77.8% (seven out of nine) of the samples from the TG or in 44.4% (four out of nine) of the samples from the CG. There were no samples exceeding the limits for counting in the TG, while there were three samples in the CG. 

### 3.3. Overall Results for the Three Farms

In 84 out of the 240 IC samples collected, *Salmonella* was present (35%; 95% IC = 29.24–41.23), and the prevalence was significantly higher in the CG than in the TG (50% vs. 20%; *p* < 0.001). 

An overall significant reduction in the proportion of *Salmonella* shedders was observed in the TG after adjusting for the effect of the farm. The odds of shedding *Salmonella* at slaughter were almost 6 times higher for the CG than for the TG (Table 4). Thus, the proportion of pigs shedding *Salmonella* that had been prevented due to the use of this OA in the TG (the AF) was 82.2%.

From the 58 samples recovered for the MPN analysis, *Salmonella* was detected (>1 CFU/g) in 15 of them (51.7%). Out of these 15, 11 (73.3%) belonged to the CG and 4 (13.8%) to the TG (*p* = 0.069). In addition, 13.8% (4/29) of the samples from the CG and 3.4% (1/29) of the samples from the TG were above the limit of counting. In general, and despite the low number of samples analyzed, a significant linear trend was observed: the higher the MPN count, the higher the odds of the sample belonging to the CG (Table 5).

## 4. Discussion

Three pig farms were selected based on serological and bacteriological evidence indicating that *Salmonella* was present. The mean serological values before the trial for these farms (21.6% in farm 1; 38.3% in farm 2; and 48.3% in farm 3) were within levels that would classify them within a category of risk, according to some national control programs [48,49,50,51]. In addition, the bacteriological study of the pens showed that *Salmonella* was present in the three farms, and this factor, along with medium–high levels of seroprevalence, has been directly related to *Salmonella* shedding at slaughter [9]. Thus, under this scenario, a considerable proportion of pigs were expected to shed *Salmonella* at slaughter. Indeed, 35% of the pigs in this study shed *Salmonella*, with this figure being even higher than that found in a previous large-scale study carried out in the same region (23.6%; [9]). Finding fattening pig farms with significant levels of seroprevalence and *Salmonella* in the pens is quite common in Spain [9,52]. Considering the risk that this situation poses for the contamination of carcasses at slaughterhouses [2,3], and the difficulties (and lack of cost-effectiveness) of the strategies in reducing the overall level of *Salmonella* infection at farms [53,54], the search for new and affordable on-farm interventions that mitigate this problem seems unavoidable.

OAs in the form of salts, and in particular formic acid, have been commonly used in livestock production to promote growth, improve feed conversion, and reduce the prevalence of specific bacterial infections [42,55,56]. Their application in esterified form is less known, but in recent years, there has been increasing interest in the potential of esterified acids to reduce the shedding of *Salmonella* in pigs [41,57]. Their enhanced antimicrobial effects against Gram-negative pathogens would arise from their diminished pH sensitivity and increased resistance to enzymatic breakdown compared to other acidic formulations. This would lead to their efficacy being observed throughout the entire gastrointestinal tract [58].

Ref. [41] showed that this esterified form of formic acid was somewhat effective when administrated for a short period of time (an average of 14 h); thus, it was expected that using the same dose (10 L/1000 L of drinking water) but increasing the time of exposure (5 days) would improve its effect.

When analyzing the results by farm, a common trend was observed; that is, on all the farms, the TG showed a lower percentage of shedders than the CG, which could be considered an indication of the positive effect of this treatment on the control of *Salmonella* shedding. On two of the farms (1 and 3), the reductions achieved were around 40% when compared to the CG (Table 2), but on farm 2, the differences were not so obvious, with a reduction of just 12.5%. However, the latter was likely a consequence of the low overall proportion of shedders from this farm. Although this farm presented a high seroprevalence when tested prior to the beginning of the trial, different unregistered factors could help to explain this situation, i.e., those associated with lower levels of pig stress. 

An analysis of the raw data suggested an overall reduction in shedding of about 30% (20% in the TG vs. 50% in the CG). However, because of the likely differences that could be found among pig farms, a statistical analysis accounting for the grouping effect of the farm was considered. According to the results of the random-effects logistic model, pigs from the CG had 5.63 (95% CI = 2.92–10.8) higher odds of shedding *Salmonella* at slaughter than pigs from the TG. This result indicates that the efficacy of this treatment to reduce the proportion of pigs shedding *Salmonella* may reach up to 82.2%. These figures are much higher than those obtained in the previous study carried out (OR = 2.8; AF = 64.3%; [41]), indicating that an increased time of exposure to the OA had a larger beneficial effect.

The amount of *Salmonella* shedding is also another parameter that should be influenced by the treatment, as OA would kill many *salmonellae* in the gut. Therefore, a lower number of them would be expected in animals from the TG. The Most Probable Number (MPN) enumeration method was used to determine whether this OA had some effect on the number of bacteria shed. This method is time-consuming and requires the analysis of the sample to be started before knowing whether it is *Salmonella*-positive. For that reason, only 10 random samples per group were considered (except for farm 1, where only 9 were used). 

As expected, and despite the limited number of samples analyzed, the number of negative ones (below the level of detection for the MPN analysis) was higher in the TG than in the CG (86.2% vs. 62.1%, respectively; *p* = 0.069). But also, among the positive samples, the load of *Salmonella* was lower in the TG. Only one sample exceeded the limits for counting (∞), and the other three samples were within the lower MPN category (0.02–20 CFU/g). By contrast, in the CG, most of the positive samples (6 out of 11) had counts above 21 CFU/g. As shown in the analysis for trends, higher MPN counts were more likely to belong to the CG (Table 5). These results support the effectiveness of this treatment to reduce the load of *Salmonella* in the pig’s gut. Since there appears to be a positive association between high cecal *Salmonella* loads in pigs and carcass contamination [4], the use of this esterified form of formic acid should be considered as a potential alternative to decrease the risk of *Salmonella* carcass contamination at slaughter.

In a previous study, it was shown that the risk, for a batch of fattening pigs, of shedding *Salmonella* at slaughter could be predicted before the pigs were sent to slaughter [9]. Therefore, this product could be applied in the drinking water of the farm, five days prior to slaughter, for batches of pigs identified as at risk of shedding *Salmonella*. It could also be included in the drinking water of the abattoir, helping to reduce the probability of shedding even more [41]. However, further studies applying different treatment protocols should be conducted to try to reduce treatment time or product dose in order to minimize the cost associated with this strategy. 

## 5. Conclusions

On *Salmonella*-positive farms, the administration to fattening pigs of this esterified formic acid through water, five days prior to slaughter, decreased the proportion of pigs shedding *Salmonella* at slaughter and the loads of *Salmonella* in the guts of pigs still shedding. Thus, this intervention may likely help to reduce abattoir and carcass contamination. These findings highlight the potential of esterified formic acid as an on-farm strategy for *Salmonella* control in pig production, being a feasible mitigation measure to be applied as a complement to other measures in the farm–slaughterhouse interface. 

## Figures and Tables

**Table 1 animals-13-02814-t001:** Bacteriological and serological results for *Salmonella* prevalence at the farms prior to trial for the control (CG) and treatment (TG) groups.

Farm	Group	No.	Seroprevalence (%)	*p*-Value	No.	Pen Prevalence (%)	*p*-Value ^1^
1	CG	30	16.67	0.53	10	10	1
	TG	30	26.70		10	10	
2	CG	30	30.00	0.29	10	10	1
	TG	30	46.67		10	10	
3	CG	30	33.33	0.04	10	10	1
	TG	30	63.33		10	10	

^1^ Fischer’s exact test, two-tailed.

**Table 2 animals-13-02814-t002:** Bacteriological results for *Salmonella* isolation from fecal samples at slaughter for the control (CG) and treatment (TG) groups.

Farm	Group	N	No. *Salmonella*-Positive Samples (%)	*p*-Value ^1^
1	CG	40	26 (65.0)	0.0015
	TG	40	11 (27.5)	
2	CG	40	5 (12.5)	0.0547
	TG	40	0 (0.0)	
3	CG	40	29 (72.5)	0.0007
	TG	40	13 (32.5)	
Total	CG	120	60 (50.0)	<0.001
	TG	120	24 (20.0)	

^1^ Fisher’s exact test, two-tailed.

**Table 3 animals-13-02814-t003:** Prevalence and concentration of *Salmonella* (MPN/g) in fecal samples at slaughter for the control (CG) and treatment (TG) groups using the MPN method.

Farm	Group	N	No. *Salmonella*-Positive Samples (%)	No. Samples in *Salmonella* MPN/g Ranges (%)			
				ND ^1^	0.02–20	21–200	∞ ^2^
1	CG	9	4 (44.4)	5 (55.6)	3 (33.3)	0 (0.0)	1 (11.1)
	TG	9	1 (11.1)	8 (88.9)	0 (0.0)	0 (0.0)	1 (11.1)
2	CG	10	1 (10.0)	9 (90.0)	1 (10.0)	0 (0.0)	0 (0.0)
	TG	10	0 (0.0)	10 (100.0)	0 (0.0)	0 (0.0)	0 (0.0)
3	CG	10	6 (60.0)	4 (40.0)	1 (10.0)	2 (20.0)	3 (30.0)
	TG	10	3 (30.0)	7 (70.0)	3 (30.0)	0 (0.0)	0 (0.0)
Total	CG	29	11 (37.9)	18 (62.1)	5 (17.2)	2 (6.9)	4 (13.8)
	TG	29	4 (13.8)	25 (86.2)	3 (10.3)	0 (0.0)	1 (3.4)

^1^ ND: not detected, below the limit of detection (<1 CFU/g). ^2^ ∞: exceeding the limit for counting.

**Table 4 animals-13-02814-t004:** Results of the random-effects logistic regression analysis * to predict *Salmonella* shedding at the abattoir.

	Coefficient (β)	Std Error (β)	Odds Ratio (OR)	95% CI (OR)	*p*-Value
Group					
Control			1	-	-
Treatment	1.73	0.33	5.63	2.92–10.8	<0.001
Constant	−1.86	0.84	0.15	0.29–080	0.026

* Farm considered as grouping (random) variable. Intraclass correlation coefficient (ICC): 0.36 (95% CI = 0.09–0.76).

**Table 5 animals-13-02814-t005:** Results for the analysis for linear trends in proportions for the control (CG) and treatment (TG) groups according to Most Probable Number (MPN) results.

MPN Category	CG (%)	TG (%)	OR *	95% CI (OR)	*p*
Not detected	18 (62.1)	25 (86.2)	1	-	-
0.02–20.00 CFU/g	5 (17.2)	3 (10.3)	2.32	0.47–11.3	0.28
21–∞ CFU/g *	6 (20.7)	1 (3.5)	8.33	0.80–8.62	0.03

*Chi*-squared for linear trend = 5.11 *p* = 0.023. * MNP categories “21–200 CFU/g” and “∞ CFU/g” were merged for the sake of the calculation (to avoid zeros).

## Data Availability

Data available upon request.

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
