# Peer review of "Salmonella Control in Fattening Pigs through the Use of Esterified Formic Acid in Drinking Water Shortly before Slaughter"

_animals, 2023, doi:10.3390/ani13182814_

Round 1

Reviewer 1 Report

The manuscript submitted by Bernard-Roche et al. confirms the efficacy of esterified formic acid administered in drinking water on Salmonella shedding at slaughter. This is a well-planned study of great interest to the sector due to the importance of pigs in the control of the second zoonosis of food origin in the European Union (salmonellosis). 

The methodology details all the analyses carried out and the results are very well presented, which makes them easy to understand. The introduction and discussion are well formulated, with current and important references in the sector.

I believe that the manuscript is ready to publish but I have a few minor comments:

L114. Experimental design. The choice of three farms and within each one two fattening units is very well explained: one for the control group and another for the treatment group. However, the number of individuals within each fattening unit is not specified. Could you please indicate the approximate number of pigs in each group?

L147. "...(except for farm 1 that only 9 were analysed)...". Why were only 9 IC samples tested on farm 1?

L159. "...a characteristic colony was biochemically tested for definitive confirmation." How was this biochemical test performed? Through API strips? Isolated biochemical tests? Please specify.

L193. "One month before starting the trials, the Salmonella status...". Please add the comma to improve the comprehension of the sentence.

L201. Table 1. Please indicate the meaning of N for each column (blood or pen samples).

Author Response

Dear reviewer, thank you very much for your comments. We have considered all of them.

L114. Experimental design. The choice of three farms and within each one two fattening units is very well explained: one for the control group and another for the treatment group. However, the number of individuals within each fattening unit is not specified. Could you please indicate the approximate number of pigs in each group?

It is now indicated that the average size per fattening unit was 1,000 animals.

L147. "...(except for farm 1 that only 9 were analysed)...". Why were only 9 IC samples tested on farm 1?

Explanation included now: “one was inadvertently forgotten”

L159. "...a characteristic colony was biochemically tested for definitive confirmation." How was this biochemical test performed? Through API strips? Isolated biochemical tests? Please specify.

Included: indole reaction, urea and triple sugar iron agar tests, and L-lysine decarboxylation.

L193. "One month before starting the trials, the Salmonella status...". Please add the comma to improve the comprehension of the sentence.

Done

L201. Table 1. Please indicate the meaning of N for each column (blood or pen samples).

Done

Reviewer 2 Report

This paper is well written with a clear objective and straight forward discussion. Also the statistics applied are accurate. 

The experimental design needs some clarification. the animals were housed in 10 pens and 30 blood samples were taken. 3 from each pen?

Did the authors look at the distribution of the seroprevalence among pens prior to the trial? . In other words would it be possible that some of the pens did not have any positive blood samples en how did this affect the bacteriological results at slaughter ?

Do I interpretate the materials and methods correctly that the TG group which were slaughtered 1 or 2 days later were also transported to the slaughterhouse later.

Could you give an indication of the distance  and or the duration of the transports from the different farms.?

Author Response

Dear reviewer, thank you very much for your comments. We have considered all your concerns as you can see in the article now:

This paper is well written with a clear objective and straight forward discussion. Also the statistics applied are accurate.

The experimental design needs some clarification. The animals were housed in 10 pens and 30 blood samples were taken. 3 from each pen?

It is clarified now in lines 122-123: … from 10 representative pens and 30 individual blood samples from pigs distributed along the fattening unit (1 pig per pen).

This was to make serological results representative of the whole fattening unit, not of the pens selected for bacteriology

Did the authors look at the distribution of the seroprevalence among pens prior to the trial? . In other words would it be possible that some of the pens did not have any positive blood samples en how did this affect the bacteriological results at slaughter ?

No, we didn´t. Serology was performed on 30 pigs selected for 30 different pens. We did not measure serology for the 10 pens used for bacteriology as we wanted to have an estimate of seroprevalence of the whole fattening unit.

We hope this issue is now more clear after including the clarification for the previous question.

Do I interpretate the materials and methods correctly that the TG group which were slaughtered 1 or 2 days later were also transported to the slaughterhouse later.

That is correct. Animals were transported the same day of slaughter.

Could you give an indication of the distance  and or the duration of the transports from the different farms?

Farms were less than 2 hours from the abattoir and all animals were transported the day of slaughter.

These clarifications have been included now in the text (lines 136-137)

Reviewer 3 Report

The manuscript by Bernard-Roche assessed the effect of esterified formic acid in drinking water in pigs on Salmonella control in abattoirs. The experimental design and statistical analysis are appropriate. The hypothesis is well explained and sound. The language is easy to understand and clear. The study comes out with interesting recommendations.

My observations are as follows-

i.          Sample size is excellent and effective to come out with a good recommendation.

ii.          L 21-24: Please provide the detail about the treatment/ methodology (formic acid concentration used in drinking water, duration, amount, etc).

iii.          Keywords: have scope for improvement, maybe formic acid in place of organic acid, microbial load, etc.

iv.          Introduction: well described the background and need for the study.

v.          L99: Plz also mentions about safety and toxicity aspects if any also economic feasability

vi.          L131: please mention as This dose (10g/kg) was below the maximum value of 12 g/kg formic acid ----

vii.          L132-133: so, during transport and lairage only water was provided without esterified formic acid.

viii.          Statistical analysis: appropriate

ix.          L281: please check the reference format

Thank you for the opportunity to read your work.

Author Response

Dear reviewer, thank you very much for your comments. We have considered all of them as indicated:

The manuscript by Bernard-Roche assessed the effect of esterified formic acid in drinking water in pigs on Salmonella control in abattoirs. The experimental design and statistical analysis are appropriate. The hypothesis is well explained and sound. The language is easy to understand and clear. The study comes out with interesting recommendations.

My observations are as follows-

i. Sample size is excellent and effective to come out with a good recommendation.   

Thank you.

ii. L 21-24: Please provide the detail about the treatment/ methodology (formic acid concentration used in drinking water, duration, amount, etc).

Done

iii.          Keywords: have scope for improvement, maybe formic acid in place of organic acid, microbial load, etc.

Done

iv. Introduction: well described the background and need for the study.

Thank you

v. L99: Plz also mentions about safety and toxicity aspects if any also economic feasibility

We did not study its economic feasibility in that previous study, only its efficacy when used at slaughter, so we cannot include that kind of comment so far. Regarding potential toxicity, as indicated in lines 130-131, the concentration used was much below the one considered of risk. We think there is no need to repeat in the introduction.

vi. L131: please mention as This dose (10g/kg) was below the maximum value of 12 g/kg formic acid ----

Done

vii.          L132-133: so, during transport and lairage only water was provided without esterified formic acid.

Transport was short and no drinking water was administered to pigs. We have included a comment on distance from farm to abattoir in lines 136-137 that qualifies the reviewer´s concern. Lairage was also short (2 hours), but water was available (without treatment).

viii.          Statistical analysis: appropriate

Thank you